# Comparison of HPLC, HPTLC, and In Silico Lipophilicity Parameters Determined for 5-Heterocyclic 2-(2,4-Dihydroxyphenyl)-1,3,4-thiadiazoles

**DOI:** 10.3390/molecules29112478

**Published:** 2024-05-24

**Authors:** Beata Paw, Rafał Śliwa, Łukasz Komsta, Bogusław Senczyna, Monika Karpińska, Joanna Matysiak

**Affiliations:** 1Department of Medicinal Chemistry, Medical University of Lublin, Jaczewskiego 4, 20-090 Lublin, Poland; beata.paw@umlub.pl (B.P.); rafal.r.sliwa@gmail.com (R.Ś.); lukasz.komsta@umlub.pl (Ł.K.); 2Department of Chemistry, University of Life Sciences in Lublin, Akademicka 15, 20-950 Lublin, Poland; boguslaw.senczyna@up.lublin.pl; 3Łukasiewicz Research Network—Institute of Industrial Organic Chemistry, Annopol 6, 03-236 Warsaw, Poland; monika.karpinska@ipo.lukasiewicz.gov.pl

**Keywords:** thiadiazoles, liquid chromatography, log k_w_, R_Mw_, log P, PCA

## Abstract

The 5-heterocyclic 2-(2,4-dihydroxyphenyl)-1,3,4-thiadiazoles were obtained as potential biologically active compounds. Lipophilicity is one of the most important physicochemical properties of compounds and was already taken into account during the drug candidates design and development. The lipophilicity of compounds was determined using the computational (log P) and chromatography (log k_w_, R_Mw_) methods. The experimental ones included the reverse-phase column high performance liquid chromatography RP (HPLC) with C8, C18, phosphatidylcholine (IAM), and cholesterol stationary phases and the thin layer chromatography (RP-HPTLC) with C8 and C18 stationary phases and various organic modifiers under the isocratic conditions. Descriptive statistics, correlation, and PCA analyses were used to compare the obtained results. For lipophilicity estimation of the tested compounds by HPTLC, dioxane and MeOH seem to be particularly beneficial as organic modifiers. The chromatographic lipophilicity parameters log k_w_ (R_Mw_) were well correlated and highly redundant (85%) compared with those calculated. Most compounds possess lipophilicity parameters within the recommended range for drug candidates.

## 1. Introduction

Lipophilicity is one of the most important physicochemical properties of drug candidates to consider during their design and development. It determines all processes in the areas of pharmacokinetics (absorption, distribution, metabolism, and excretion (ADME)) as well as toxicity and influences the pharmacodynamic phenomena [1,2]. Lipophilicity expresses the ability of a compound to dissolve in fats, lipids, and other non-polar organic solvents such as octane-1-ol or toluene [3]. It is usually expressed as log P calculated as the decimal logarithm from the ratio of analyte concentrations in the organic phase (octane-1-ol) and in the aqueous phase in the equilibrium state after the division between both phases [4,5,6]. This feature of a compound should be within a certain range, which enables a compound both to dissolve in water and to penetrate passively phospholipid membranes. Both of these processes require different properties from a compound [4]. It is assumed that the molecules should have a balanced lipophilic–hydrophilic character expressed by the partitioning coefficient log P in the range of 0–3 and by the distribution coefficient log D_7.4_ in the range of 1–3 [3]. According to Lipinski, taking numerical methods into account, the limit of Clog P is less than five [7,8,9]. Compounds that are too lipophilic dissolve very poorly in water, which confines their pharmacokinetic processes and may cause accumulation in fatty phases. On the other hand, compounds with low lipophilicity cannot cross the biological membrane barrier by passive diffusion. Many properties such as anticancer [10,11] and antifungal [12] activities or toxicity are simply related to lipophilicity. Therefore, these descriptors are taken into account at the earliest possible stage of research on a drug candidate to eliminate unpromising molecules, shorten the development time, and reduce costs [3,4,13].

There were two groups of methods developed for the determination of compound lipophilicity: computational and experimental ones. Among the experimental methods, apart from the very important and classic shake flask method, liquid chromatography is the most frequently applied [14]. For these purposes, various chromatographic methods were developed, including high performance thin layer chromatography (HPTLC) [15,16,17,18], column high performance liquid chromatography (HPLC) [19], over-pressured layer chromatography (OPLC) [20,21], micellar electrokinetic chromatography (MEKC) [4,15,16], and others. The chromatographic RP-HPLC method is recommended by the International Union of Pure and Applied Chemistry (IUPAC) as an equivalent one to the shake flask technique for log P estimation [19].

For the lipophilicity assessment of the compounds, the octadecyl C18 stationary phase and the less hydrophobic octyl C8 one in the reversed phase (RP) system are the most frequently used [4,19,22]. Both columns with such fillings and TLC plates are commercially available and commonly applied. The retention process in such systems is determined by the partition phenomena based on hydrophobic interactions.

In column chromatography, phase affinity is also assessed using immobilized artificial membrane (IAM) [19,21] or cholesterol phases (Chol) [23,24]. The IAM phase contains chemical bonding of phosphatidylcholine to a solid surface of silica gel, and this appears to be a better model for biological membranes than the C8 and C18 phases. Apart from hydrophobic interactions, retention is determined by ionic interactions with the ionized groups of the bound amino alcohol [14,15]. The effects are particularly visible in the case of polar or ionized compounds.

Mainly MeOH and ACN are used as organic modifiers in the reverse phase system (RP). Other organic modifiers such as EtOH, acetone [17,25], THF, and dioxane [26] were also considered, especially in HPTLC chromatography. However, due to it having the greatest similarity to water, MeOH seems to be the best solvent in many cases [22,27].

The 5-heterocyclic 2-(2,4-dihydroxyphenyl)-1,3,4-thiadiazoles were prepared as potentially biologically active compounds. The 5-aryl/(arylamino) and alkyl/alkylamino 2-(2,4-dihydroxyphenyl)-1,3,4-thiadiazoles previously described by our team exhibit anticancer and antifungal properties [28]. Some of them are cholinesterases inhibitors with the activity at nM level (IC_50_) [29]. Expanding the research in this area, a synthesis was developed and accomplished, giving 5-heterocyclic 2-(2,4-dihydroxyphenyl)-1,3,4-thiadiazole derivatives, which are the subject of the research in the paper. The selected compounds are cholinesterases inhibitors [30]. 

The aim of the research was to assess and compare the lipophilicity of the newly prepared 1,3,4-thiadiazoles using various chromatographic systems as well as different computational algorithms. The lipophilicity parameters obtained from the diversified systems were compared using the chemometric methods to find similarities and differences in the behavior of compounds in the individual chromatographic systems. The lipophilicity amounts of the compounds under consideration were analyzed in terms of drug-likeness to justify further research on their biological activities and toxicity profiles.

## 2. Results and Discussion

### 2.1. A Set of the Compounds

The tested compounds are presented in Table 1. These are 5-heterocyclic 2-(2,4-dihydroxyphenyl)-1,3,4-thiadiazoles additionally modified with lipophilic Et or Cl substituents in the resorcinol ring. Heterocyclic systems are six-membered and five-membered rings as well as five-membered rings fused with a six-membered one. Some of them are additionally substituted with phenyl groups that increase the lipophilicity of these types of compounds.

### 2.2. Chromatographic Lipophilicity Parameters

Chromatographic studies were carried out using both HPTLC and HPLC methods under the isocratic conditions. In all systems, there were stated linear relationships between the R_M_ (HPTLC) or log k_w_ (HPLC) values and the organic modifier content (ϕ) in the mobile phase. This is described by the Soczewinski–Wachtmeister equation [31] as follows:R_M_ = R_Mw_ + bϕ(1)
log k = log k_w_ + bϕ(2)
where ϕ expresses the volume fraction of the organic solvent in the mobile phase; R_Mw_ (log k_w_) corresponds to 0% of the organic solvent content in the mobile phase; b represents the slope of a regression line [11]. 

The retention behavior of the compounds was analyzed by HPTLC using the octyl C8 and octadecyl C18 stationary phases and MeOH, ACN, acetone, and dioxane as organic modifiers (Appendix A). The chromatographic evaluation of using the HPLC methods was made in the systems, with C8/MeOH, C8/ACN, and C18/ACN analogous to HPTLC (Appendix A). The measurements using the IAM and Chol stationary phases and the buffered mobile phases (pH = 7.4) were additionally used (Appendix A).

### 2.3. Structure–Retention Relationship

Analyzing the lipophilicity and the structure of the compounds, it was found that the derivatives with an additionally unsubstituted benzenediol ring were characterized by lower lipophilicity compared to the analogues containing the Et or Cl substituent (compounds **4**, **6**, **11**, **13**). The lipophilicity parameters for the Et derivatives were usually higher than those for chlorine ones (**1** and **2**; **7** and **8**). This was an expected effect, taking into account the hydrophobicity parameter π (Hansch–Fujita π constant) for these substituents [32]. Deviations from this order were sometimes observed. Taking into account the structure of the heterocyclic substituent, compound **13** with the 6-(morpholin-4-yl)pyridin-3-yl-substituent was characterized by the lowest lipophilicity. Furanyl derivative (**1**) (even though it contained a Cl substituent) and compound **6** with the 2,1,3-benzoxadiazol-5-yl substituent were also characterized by low lipophilicity. The compounds containing two phenyl groups in the heterocyclic ring possessed the highest lipophilicity (**15**–**17**). (Appendix A).

### 2.4. Descriptive Statistics of Lipophilicity Parameters

To compare the obtained lipophilicity parameters, first, the selected parameters of the descriptive statistics of log k_w_ (R_Mw_) were taken into account (Table 2). 

The retention mean values R_Mw_ from the highest to the lowest were presented as follows: MeOH > dioxane > acetone > ACN obtained using both C8 and C18 stationary phases. For log k_w_ using C8, the results were: MeOH > ACN. Taking into account the stationary phases, the R_Mw_ values using C18 were higher than they were when using C8. For HPLC, the log k_w_ values were presented in the following order: Chol > C18 > IAM. The R_Mw_ parameters covered a larger range for MeOH and dioxane, while the smallest range was observed for ACN as an organic modifier. In the case of the C18 HPLC system, it was also larger for MeOH than for ACN. The greatest diversity of R_Mw_ was observed in the C8/dioxane and C18/dioxane systems, followed by the MeOH ones, and the smallest diversity was presented in the ACN ones. In the case of HPLC chromatography, the greatest and the smallest diversities were observed for the C18/ACN and IAM systems, respectively. Higher log k_w_ values in C18 than in the IAM phase were observed for various groups of compounds [19,30,33]. RP-18 HPLC studies gave similar results: the highest R_Mw_ values for the MeOH mobile phase and the lowest with ACN [34]. However, in the other C8 HPTLC studies involving 2,4-dihydroxythiobenzanilides, the lowest R_Mw_ values were obtained with MeOH used as an organic modifier compared to acetone and ACN [35]. This could indicate that, in this aspect, not only the type of organic modifier but the structures of the compounds are important.

### 2.5. Correlation Analysis

Table 3 presents the correlation matrix between the R_Mw_ and log k_w_ parameters obtained using different chromatographic systems.

The highest correlation was found for the R_Mw_ parameters obtained from the C18/dioxane and C8/acetone systems and is expressed by the following equation:R_Mw_(C18/diox) = 0.1194 (±0.2381) + 1.0845 (±0.0636) R_Mw_(C8/acet)(3)
$$n=18,\;R=0.9736,\;R^{2}=0.9478,\;R_{adj}^{2}=0.9446,\;F(1,16)\;=\;290.62,\;p\;<\;0.0000,\;s\;=\;0.2890$$

The statistical parameters indicate that the numerical values of R_Mw_ for individual compounds were similar to each other. This was also confirmed by the parameters of descriptive statistics. The next equation includes the lipophilicity parameters obtained using the C18 stationary phase with dioxane and acetone as organic modifiers, as follows:R_Mw_(C18/diox) = −0.4258 (±0.3079) + 1.1687(±0.0772) R_Mw_(C18/acet)(4)
$$n=18,\;R=0.9668,\;R^{2}=0.9347,\;R_{adj}^{2}=0.9306,\;F(1,16)\;=\;229.09,\;p\;<\;0.0000,\;s\;=\;0.3233$$

Equations (3) and (4) show that the R_Mw_ parameters obtained from the C18/dioxane system were better correlated and less differentiated with the R_Mw_ values from C8/acetone than with R_Mw_ from the C18/acetone system. 

A high correlation in C18 phases was also found for R_Mw_ values obtained with ACN and dioxane as organic modifiers, as follows:R_Mw_(C18/diox) = 0.4312 (±0.3125) + 1.4799 (±0.1232) R_Mw_(C18/CAN)(5)
$$n=18,\;R=0.9488,\;R^{2}=0.9001,\;R_{adj}^{2}=0.8939,\;F(1,16)\;=\;144.22,\;p\;<\;0.0000,\;s\;=\;0.3999$$

The lowest correlation in this stationary phase was found for the parameters obtained with MeOH and ACN as organic modifiers (r = 0.8743).

In the case of the C8 phase, the highest correlations were found for the R_Mw_ parameters obtained for acetone-ACN and acetone-MeOH as organic modifiers. They are expressed by Equations (6) and (7), as follows:R_Mw_(C8/acet) = 0.04311 (±0.3103) + 1.2539 (±0.1055) R_Mw_(C8/ACN)(6)
$$n=18,\;R=0.9477,\;R2^{\;}=0.8992,\;R_{adj}^{2}=0.8918,\;F(1,16)\;=\;141.13,\;p\;<\;0.0000,\;s\;=\;0.3625$$


R_Mw_(C8/acet) = −0.3687 (±0.3625) + 0.95702 (±0.0850) R_Mw_(C8/Met)(7)

$$n=18,\;R=0.9423,\;R^{2}=0.8880,\;R_{adj}^{2}=0.8810,\;F(1,16)\;=\;126.84,\;p\;<\;0.0000,\;s\;=\;0.3801$$



Similarly to the C18 phase, the lowest correlation was found for the pair of organic modifiers MeOH-ACN (r = 0.8987) using the C8 stationary phase.

The relationships between the lipophilicity parameters log k_w_ obtained by HPLC and the stationary phases IAM, Chol, and C18 are described by the following equations:log k_w_(IAM) = 0.01642 (±0.2803) + 0.63568 (±0.0677) log k_w_(C18)(8)
$$n=18,\;R=0.9200,\;R^{2}=0.8464,\;R_{adj}^{2}=0.8368,\;F(1,16)\;=\;88.15,\;p\;<\;0.0000,\;s\;=\;0.2949$$


log k_w_(Chol) = 0.8608 (±0.3457) + 0.88990 (±0.0835) log k_w_(C18)(9)

$$n=18,\;R=0.9362,\;R^{2}=0.8765,\;R_{adj}^{2}=0.8688,\;F(1,16)\;=\;113.58,\;p\;<\;0.0000,\;s\;=\;0.3637$$




log k_w_ (IAM) = −0.3050 (±0.3734) + 0.64808 (±0.0823) log k_w_(Chol)(10)

$$n=18,\;R=0.8915,\;R^{2}=0.7948,\;R_{adj}^{2}=0.7820,\;F(1,16)\;=\;61.98,\;p\;<\;0.0000,\;s\;=\;0.3408$$



The lipophilicity parameters log k_w_ obtained using the C18 and Chol stationary phases correlated well. The lowest correlation was found for the Chol-IAM systems.

Comparing the same C18/ACN, C8/MeOH, and C8/ACN chromatographic systems and the different methods, the following correlation coefficients were found: 0.9043, 0.9201, and 0.8943, respectively. Additionally, they were higher for MeOH than for ACN as organic modifiers. The best system is described by the following equation:log k_w_(C8/MeOH) = 1.5958 (±0.3753) + 0.82654 (±0.0880) R_Mw_(C8/MeOH)(11)
$$n=18,\;R=0.9201,\;R^{2}=0.8465,\;R_{adj}^{2}=0.8369,\;F(1,16)\;=\;88.26,\;p\;<\;0.0000,\;s\;=\;0.3936$$

For comparison, the log P and log D values were calculated using various approaches and computational software. Marvin Sketch software was included in the studies because of the ability to calculate log D depending on pH [36]. The results are presented in Table 4.

The lipophilicity level of the compounds under consideration expressed as log P was analyzed with regard to their drug-likeness. Compounds **5**, **16**–**18** possess the predicted Clog P values higher than 5 and do not meet one of the Lipinski rules of five [4,5,35]. Some of them, the most lipophilic ones (compounds **16**, **17**), do not meet the Oprea recommendations. Their Moriguchi log P (Mlog P) values are not in the range from −2.0 to 4 [6]. Phenyl groups are important in biologically active molecules due to potential hydrophobic interactions with the hydrophobic pocket of the molecular target or π-π interactions. Two substituents of this type, in the compounds under consideration, especially chlorinated, increase lipophilicity significantly and can yield derivatives beyond the recommended lipophilicity range.

The correlation matrix (r) of R_Mw_ or log k_w_ parameters obtained by HPTLC or HPLC, respectively using various stationary phases and the log P (log D) parameters calculated using various computational approaches are presented in Table 5.

The obtained results indicated that the calculated Clog P, AlOGP, log P, and log D_7.4_ descriptors correlated the best with the chromatographic data. The R_Mw_ values from the C8/dioxane system were particularly highly correlated (r = 0.97). The other chromatographic system from which the lipophilicity parameters correlated well with the calculated ones was C8/acet HPTLC. The lowest correlations were found for the IAM HPLC and the C18/MeOH and C8/MeOH HPTLC systems. There were better correlations of log P found with the log k_w_ values obtained using the Chol phase compared to using the IAM one. No satisfactory correlations of log D_7.4_ and the parameters obtained from the buffered systems (pH = 7.4) were found. However, it is worth paying attention to the fact that chromatographic descriptors are generally highly correlated with the calculated log P values. 

Additionally, the above two data sets (slopes and intercepts) were analyzed together with the computational descriptors by means of the hierarchical cluster analysis using the correlation coefficient as a dissimilarity measure. The resulting dendrogram is presented in Figure 1. It can be concluded that the worst correlated parameters were the slope of the equation fitted to the HPLC system with the C8 column and the ACN modifier as well as that of the IAM equation.

The MeOH systems formed a strictly correlated cluster (T-C18-MET-S, T-C18-MET-R, C-C8-MET-S, C-C8-MET-R), whereas the other systems were less correlated with the MeOH ones. The R_Mw_ values of the IAM equations were similar to the thin-layer systems on the RP8 plates. The CHOL-S descriptor seemed to be the one possessing the best correlation with the computational ones.

The increase in the chromatographic lipophilicity parameters of the tested compounds was not the same in all systems, consistent with the theoretically calculated log P or log D parameters. This should be explained by the different solvation effects of various solvents, tautomeric processes—mainly heterocyclic systems (also dependent on the type of solvent)—or the influence of pH on the structure of the analyte [40]. It can be assumed that experimental methods are also necessary while investigating lipophilicity and chromatographic behavior, and there is unstrict correlation between the experimental R_MW_ and the computational lipophilicities. 

The research proves that, in the case of the tested group of compounds, the log k_w_ parameters obtained using the IAM phase constitute a set of slightly different data than those using alkyl phases. The presence of polar benzenediol groups and various heterocyclic rings interacting with the ionized fragments of phospholipids of the IAM phase resulted in different interactions, which translated to the retention as well as the size and the selectivity of the log k_w_ parameter. 

### 2.6. Principal Component Analysis

To investigate the similarity between the investigated compounds, the obtained experimental and computational data were analyzed using scaled principal component analysis. The results are presented in Figure 2 and Figure 3. 

The most important conclusion from the PCA analysis is that all coefficients, both computational and experimental ones, were highly redundant. In total, 85% of the information was common for them and could be collected together to form the first principal component. The score could be interpreted as an average value of lipophilicity. The compounds with a high value of PC (compounds **5** and **15**–**18**) had the highest lipophilicity, whereas the low score values corresponded to the lowest lipophilicity (compounds **19**, **9**, etc.).

The second principal component was the main component modeling differences between chromatographic systems. However, only 3.6% of the information could be placed into this PC, which represents the following trend: compounds with a high score value (compounds **1**, **12**, **14**, **18**) had high R_MW_ values in IAM and C8 HPLC with MeOH as well as the high slope of this equation compared to all other systems. A low score value meant the opposite trend (compounds **5**, **6**, **8**, **10**). 

The third principal component represented the second differentiation trend concerning MeOH and ACN with a similar variance (3.4%). The low score of this PC (compounds **7**, **9**, **10**, **18**) indicated the low R_MW_ and the low slope (S) in the MeOH systems with correspondingly high values using the ACN systems. The high scores (**3**, **11**, **12**) indicated the high R_MW_ and a high slope with MeOH and a low slope with ACN, which was the opposite behavior. 

## 3. Materials and Methods

### 3.1. Compounds

Compounds under consideration (compounds **1**–**18**) were synthesized as biologically active agents. They were prepared by treatment of commercially available hydrazides with sulfinylbis[(2,4-dihydroxyphenyl)methanethione] (STB) or its analogues [28]. Some compounds were described as acetylcholinesterase (AChE) and butyrylcholinesterase (BuChE) inhibitors [30]. 

### 3.2. HPTLC Chromatography

Methanol solutions of substances with a concentration of 1 mg/mL were used in the tests. In total, 2 µL of solutions of the tested substances were placed on the RP-18 F_254_ and RP-8 F_254_ chromatographic plates (10 × 20 cm) at the starting line marked at a distance of 1 cm from the lower edge of the longer edge of the plate. The plates with applied solutions of the compounds were developed in a horizontal chromatographic chamber (DS L, Chromdes Lublin, Poland) at a distance of 9 cm. Mobile phases containing water and MeOH, ACN, acetone, or dioxane as organic modifiers were used. The organic solvent contents in the mobile phases were as follows: RP-18 F_254_ phase: MeOH: 60–90%; acetonitrile and acetone: 50–90%; 1,4–dioxane: 40–80%; RP-8 F_254_ phase: MeOH: 50–90%; acetonitrile: 45–70%; acetone and 1,4-dioxane: 50–80%. The spectral grade solvents used in the experiment were purchased from Merck, Darmstadt (Germany).

After development of the chromatograms, the plates were dried at room temperature. Next, spots were visualized at 254 nm and 366 nm under the analytical UV lamp (Haaland HA-05, Warsaw, Poland). R_F_ values (retention factors) were calculated from the equation as follows:(12)$$R_{F}=\frac{a}{b}$$
where: a—the migration distance of a compound (mm); b—the migration distance of a mobile phase (mm). The R_F_ values obtained in this way were transformed into R_M_ values, which are the equivalent of the log k parameter of column chromatography. They were calculated based on the Bate-Smith and Westall Equation (13) [41]:R_M_ = log[(1/R_F_) − 1](13)

### 3.3. HPLC Chromatography

HPLC measurements were performed using a liquid chromatograph Knauer (Knauer, Berlin, Germany) with a dual pump, a 20 µL simple injection valve, and a UV-visible detector. Measurements were carried out at room temperature. The compounds were detected under UV light at 254, 280, or 330 nm. The retention time of an unretained solute (t_0_) was determined by the injection of a small amount of citric acid dissolved in water. The spectral grade solvents used in the experiment were purchased from Merck, Darmstadt (Germany).

In the RP-8 HPLC chromatography, the Symmetry C-8 (100 Å, 5 μm, 250 × 4.6 mm) Waters Corporation (Milford, NC, USA) column was used as a stationary phase. The MeOH and ACN concentrations in the mobile aqueous phase ranged from 0.6 to 0.9 and from 0.35 to 0.9 (*v*/*v*), respectively, at 0.05 intervals. The flow rate was 1 mL × min^−1^ (Appendix A).

In the RP-18 HPLC chromatography process, the Eurosil Bioselect C-18 (100 Å, 5 μm, 300 × 4.0 mm) column was used as the stationary phase. The mobile phase included different volume mixtures of MeOH as the organic modifier and 20 mM acetate buffer as the aqueous phase to obtain pH = 7.4. The MeOH concentration ranged from 0.4 to 0.9 (*v*/*v*) at 0.1 intervals (Appendix A). 

In the IAM HPLC chromatography, a Rexchrom IAM.PC.DD2 (300 Å, 12 μm, 100 × 4.6 mm) (Regis Technologies, Morton Grove, IL, USA) column was used as the stationary phase. The compounds were dissolved at a concentration of 0.5 mg × mL^−1^ in MeOH. The mobile phases consisted of different volume fractions of ACN and 20 mM phosphate buffer; the aqueous phase maintained a pH = 7.4 (0.02 M KH_2_PO_4_, Na_2_HPO_4,_ and 0.15 M KCl). The ACN concentration ranged from 0.05 to 0.4 (*v*/*v*), depending on the structure of the compound, at 0.05 intervals (Appendix A). The flow rate was 1 mL × min^−1^. 

The cogent 4 UDC Cholesterol (100 Å, 4 µm, 150 × 2.1 mm) MicroSolv Technology Corporation (Leland, NC, USA) column was used. The mobile phase included different volume mixtures of MeOH as the organic modifier and 20 mM acetate buffer as the aqueous phase to obtain pH = 7.4. The concentrations of the organic modifier were in ranges from 0.3 to 0.8 (*v*/*v*) with steps of 0.05 or 0.1 (Appendix A). The flow rate was 0.35 mL × min^−1^.

### 3.4. Computational Methods

CA log P, Slog P, and log D were calculated using the Marvin Sketch ver. 19.9 software [36]. The ChemAxon model CA log P based on the VG method was used [42]. A consensus model, Con log P, was based on the Chemaxon, Klopman et al. models and the PhysProp database [43]. Using the ALOGPS ver. 2.1 software, ALOGP values were calculated [37]. The clog P and log P values were estimated using the ChemDraw Ultra ver. 10.0 [38] according to the fragmentation method introduced by Crippen [44]. The Moriguchi Mlog P, S + log D, and S + log P were estimated by the MedChem Designer (TM) v. 3.0.0.30 [39]. Statistica version 7.1 was used for the regression and correlation analyses [45]. The principal component analysis and the hierarchical cluster analysis were performed in R (version 4.3.3) using the built-in functions for the computation and “ggplot2” package for plot generation.

## 4. Conclusions

The chromatographic lipophilicity parameters log k_w_ (R_Mw_) were generally well correlated with the calculated partition coefficients log P (log D). They were highly redundant, and 85% of the information was common. The correlations of those obtained from the C18/dioxane HPTLC system with the calculated Clog P, AlOGP, log P, and log D_7.4_ were particularly high. Additionally, the S descriptor obtained using the Chol phase seemed to possess the best correlation with the computational ones. The lowest correlations were found for those obtained using the IAM phase. 

The research proves that, in estimating lipophilicity of the tested compounds using planar chromatography, dioxane and MeOH seem to be particularly beneficial as organic modifiers. The obtained lipophilicity parameters cover the widest range being characterized by the greatest diversity. Dioxan also allows the use of the largest amount of water in the mobile phase, which can affect the results using the extrapolation method. The best correlation here was between the R_Mw_ parameters obtained using dioxane and acetone as organic modifiers.

As follows from the research, the log k_w_ values obtained using the IAM stationary phase, in the case of the tested group of compounds, constitute sets of slightly different information compared with those obtained using alkyl phases. Therefore, log k_w_ (IAM) can be considered as another descriptor in the potential structure–activity studies in this group of analogues.

The lipophilicity of the newly prepared compounds is within the range recommended for potential drug candidates, for which favorable pharmacokinetic processes, especially absorption and distribution after oral administration, are predicted. Therefore, the selected promising compounds will be subjected to a wide range of anticancer studies. 

## Figures and Tables

**Figure 1 molecules-29-02478-f001:**
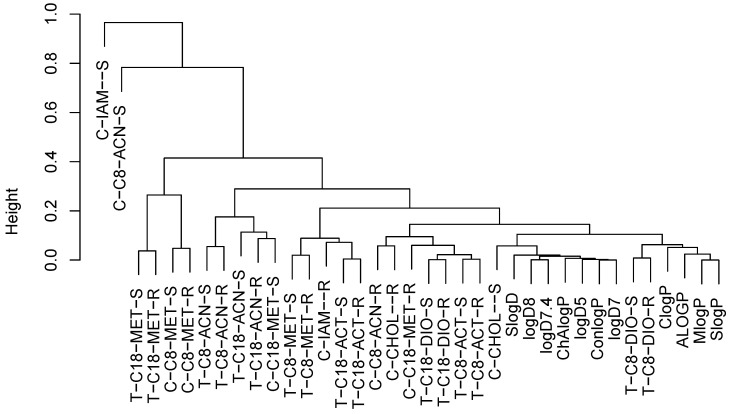
Hierarchical cluster analysis of chromatographic systems and computational descriptors using correlation as a dissimilarity measure. The abbreviations are created from the system type (T—HPTLC, C—HPLC), the stationary phase, and a coefficient (R—R_Mw_, S—slope).

**Figure 2 molecules-29-02478-f002:**
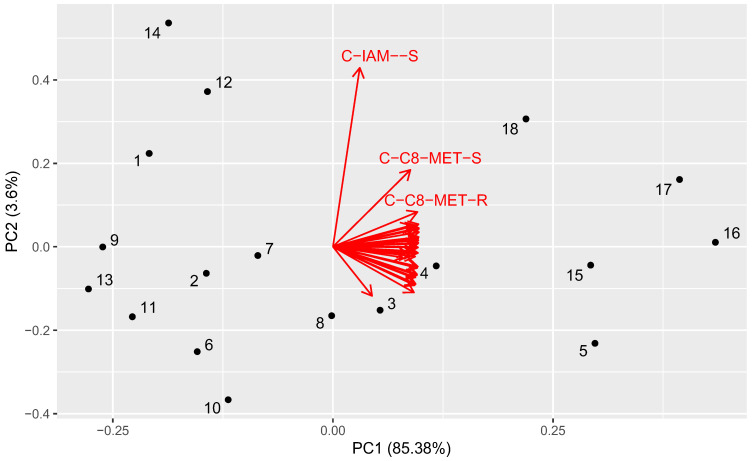
The first and second principal components of the analyzed dataset. The numbers correspond to the compounds. The red arrows denote the loading vectors of corresponding original matrix columns. The abbreviations are created from the system type (T—HPTLC, C—HPLC), the stationary phase, and a coefficient (R—R_Mw_, S—slope).

**Figure 3 molecules-29-02478-f003:**
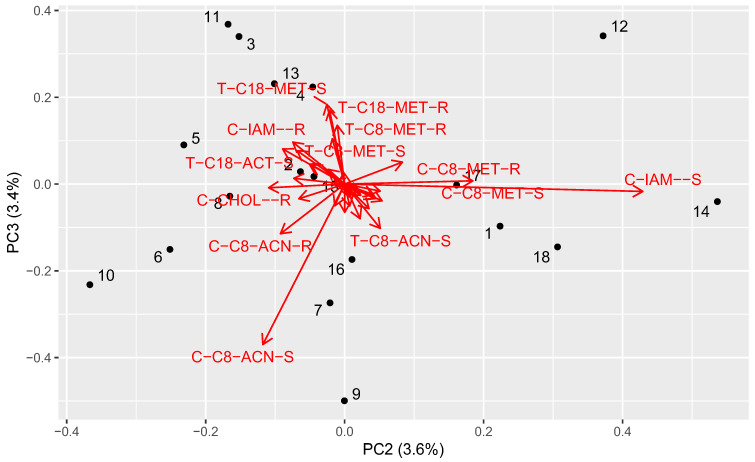
The second and third principal components of the analyzed dataset. The numbers correspond to the compounds. The red arrows denote the loading vectors of corresponding original matrix columns. The abbreviations are created from the system type (T—HPTLC, C—HPLC), the stationary phase, and a coefficient (R—R_Mw_, S—slope).

**Table 1 molecules-29-02478-t001:** Structure of 5-heterocyclic 2-(2,4-dihydroxyphenyl)-1,3,4-thiadiazoles.

	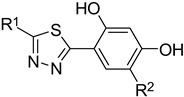
No.	R^1^	R^2^	No.	R^1^	R^2^
1.	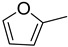	-Cl	11.	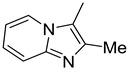	-H
2.	-Et	12.	-Cl

3.	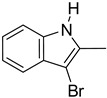	-H	13.14.	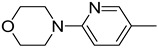	-H-Cl
4.	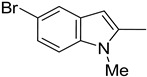	-H		
5.	-Et	15.	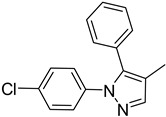	-H
6.	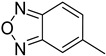	-H	16.	-Et
7.	-Cl			
8.	-Et	17.	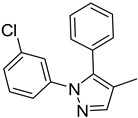	-Et
9. 10.	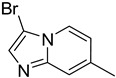	-H-Cl	18.	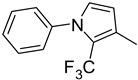	-Cl

**Table 2 molecules-29-02478-t002:** Comparison of R_Mw_ and log k_w_ values of the investigated compounds statistics in various chromatographic systems.

No.	Descriptor/System	Mean	Median	Min	Max	Range	Std. Dev.
1.	R_Mw_—C18/MeOH	4.3424	4.1679	2.5407	6.3131	3.7724	1.0561
2.	R_Mw_—C18/ACN	2.4181	2.2442	1.3930	3.9912	2.5982	0.7871
3.	R_Mw_—C18/acet	3.7951	3.2917	2.2885	5.6072	3.3187	1.0156
4.	R_Mw_—C18/diox	4.0096	3.7405	2.1989	6.1654	3.9665	1.2277
5.	R_Mw_—C8/MeOH	4.1334	3.8175	2.2257	6.5014	4.2757	1.0851
6.	R_Mw_—C8/ACN	2.8262	2.5901	1.7456	4.7899	3.0443	0.8329
7.	R_Mw_—C8/acet	3.5870	3.3786	2.1722	5.9450	3.7728	1.1021
8.	R_Mw_—C8/diox	3.8056	3.2432	1.8586	6.4537	4.5951	1.2498
9.	log k_w_—C18/MeOH	4.0104	3.9980	2.5789	5.8533	3.2744	1.0563
10.	log k_w_—C8/MeOH	5.0122	4.6529	3.5488	6.5561	3.0073	0.9748
11.	log k_w_—C8/ACN	2.3201	2.3214	1.4122	3.3937	1.9815	0.5503
12.	log k_w_—IAM	2.5658	2.4026	1.2283	3.8012	2.5729	0.7299
13.	log k_w_—Chol	4.4297	4.1548	3.1313	6.1235	2.9922	1.0041

**Table 3 molecules-29-02478-t003:** Correlation matrix (r) of R_Mw_ and log k_w_ parameters obtained by HPTLC and HPLC techniques, respectively.

Descriptor/ System	C18/MeOH	C18/ACN	C18/Acet	C18/Diox	C8/MeOH	C8/ACN	C8/Acet	C8/Diox	C18/ACN	C8/MeOH	C8/ACN	IAM	Chol
R_Mw_—C18/MeOH	1.00												
R_Mw_—C18/ACN	0.80	1.00											
R_Mw_—C18/acet	0.87	0.93	1.00										
R_Mw_—C18/diox	0.89	0.95	0.97	1.00									
R_Mw_—C8/MeOH	0.88	0.89	0.94	0.92	1.00								
R_Mw_—C8/ACN	0.79	0.94	0.90	0.91	0.90	1.00							
R_Mw_—C8/acet	0.88	0.93	0.95	0.97	0.94	0.95	1.00						
R_Mw_—C8/diox	0.81	0.91	0.91	0.91	0.88	0.93	0.94	1.00					
log k_w_—C18/MeOH	0.87	0.90	0.92	0.95	0.89	0.86	0.94	0.92	1.00				
log k_w_—C8/MeOH	0.89	0.91	0.90	0.92	0.92	0.92	0.94	0.94	0.90	1.00			
log k_w_—C8/ACN	0.76	0.90	0.88	0.91	0.82	0.89	0.92	0.92	0.92	0.86	1.00		
log k_w_—IAM	0.88	0.86	0.94	0.94	0.95	0.85	0.93	0.84	0.92	0.87	0.86	1.00	
log k_w_—Chol	0.82	0.90	0.91	0.92	0.85	0.88	0.91	0.90	0.94	0.88	0.94	0.89	1.00

**Table 4 molecules-29-02478-t004:** Log P (log D) values predicted in silico using various calculation algorithms.

No.	ChA log P ^1^	Con log P ^1^	ALOGP ^2^	log P ^3^	CLog P ^3^	Mlog P ^4^	S + Log P ^4^	S + Log D ^4^	log D_5_ ^1^	log D_7_ ^1^	log D_8_ ^1^	log D_7.4_ ^1^
1.	2.35	2.85	2.98	3.03	2.3173	1.664	1.664	2.596	2.85	2.72	2.13	2.366
2.	2.69	3.21	3.02	3.38	2.9049	1.928	1.928	3.108	3.21	3.19	3.08	3.124
3.	3.54	3.98	4.27	4.28	2.6716	2.647	2.647	4.009	3.98	3.95	3.73	3.818
4.	3.79	4.20	4.02	4.52	4.1351	2.882	2.882	4.005	4.2	4.17	3.96	4.044
5.	4.65	5.16	4.95	5.42	5.1131	3.337	3.337	4.590	5.16	5.15	5.03	5.078
6.	2.51	2.54	2.81	- ^5^	2.1569	1.871	1.871	2.909	2.54	2.51	2.30	2.384
7.	3.03	3.14	3.51	- ^5^	2.5473	2.127	2.127	2.770	3.14	3.02	2.46	2.684
8.	3.37	3.50	3.36	- ^5^	3.1349	2.377	2.377	3.467	3.50	3.48	3.38	3.42
9.	2.31	2.45	3.06	3.42	2.6964	2.040	2.040	2.913	2.21	2.41	2.20	2.284
10.	2.83	3.05	3.81	3.98	3.0865	2.281	2.281	2.953	2.81	2.92	2.36	2.584
11.	1.81	2.03	2.87	3.48	2.2487	1.918	1.918	2.481	1.98	2.00	1.78	1.868
12.	2.33	2.63	3.72	4.03	2.6387	2.161	2.161	2.736	2.49	2.53	1.93	2.17
13.	2.19	2.46	2.34	3.12	1.2077	1.552	1.552	2.403	2.12	2.42	2.21	2.294
14.	2.71	3.06	2.96	3.68	1.5977	1.789	1.789	2.895	2.72	2.93	2.37	2.594
15.	4.80	5.45	5.14	6.14	4.9270	3.850	3.850	4.692	5.45	5.42	5.20	5.288
16.	5.67	6.41	5.71	7.05	5.9050	4.262	4.262	5.167	6.41	6.39	6.28	6.324
17.	5.67	6.41	5.72	7.05	5.9050	4.262	4.262	5.201	6.41	6.39	6.28	6.324
18.	4.98	5.55	5.22	5.78	5.0844	3.485	3.485	4.832	5.55	5.42	4.85	5.078

^1^—log P and log D models prediction by Marvin Sketch ver. 19.9 software [36]. ^2^—log P model prediction by ALOGPS ver. 2.1 [37]. ^3^—log P models calculated by ChemDraw Ultra 10.0. software [38]. ^4^—log P calculated by MedChem Designer 3.0.0.30 tools [39]. ^5^—log P values were not calculated.

**Table 5 molecules-29-02478-t005:** The correlation matrix (r) of R_Mw_ or log k_w_ parameters obtained by HPTLC or HPLC chromatography, respectively using various stationary phases and the log P (log D) parameters calculated using various computational approaches.

Descriptor /System	ChA log P ^1^	Con log P ^1^	ALOGP ^2^	log P ^3^	Clog P ^3^	M log P ^4^	S + Log P ^4^	S + log D ^4^	log D_5_ ^1^	log D_7_ ^1^	log D_7.4_ ^1^
R_Mw_											
C18/MeOH	0.8318	0.7745	0.8512	0.8512	0.8290	0.8138	0.8057	0.7993	0.8030	0.8318	0.8512
C18/ACN	0.8779	0.9018	0.8977	0.8977	0.9105	0.9110	0.9064	0.9186	0.9153	0.8779	0.8977
C18/acet	0.9154	0.8946	0.9325	0.9325	0.9199	0.9320	0.9318	0.9403	0.9385	0.9154	0.9325
C18/diox	0.9308	0.9094	0.9370	0.9370	0.9180	0.9262	0.9285	0.9302	0.9310	0.9308	0.9370
C8/MeOH	0.8733	0.8234	0.8792	0.8792	0.8859	0.8967	0.8951	0.9014	0.9004	0.8733	0.8792
C8/ACN	0.9351	0.8843	0.9332	0.9332	0.9227	0.9316	0.9365	0.9439	0.9425	0.9351	0.9332
C8/acet	0.9560	0.9148	0.9544	0.9544	0.9183	0.9297	0.9318	0.9267	0.9301	0.9560	0.9544
C8/diox	0.9602	0.9544	0.9732	0.9732	0.9700	0.9542	0.9523	0.9453	0.9495	0.9602	0.9732
log k_w_											
C18/MeOH	0.8933	0.9199	0.9210	0.9210	0.9069	0.8861	0.8856	0.8839	0.8860	0.8933	0.9210
C8/MeOH	0.9447	0.9450	0.9545	0.9311	0.9224	0.9456	0.9456	0.9599	0.9531	0.9454	0.9385
C8/ACN	0.9262	0.9194	0.9315	0.9059	0.9513	0.9425	0.9425	0.9308	0.9236	0.9232	0.9276
IAM	0.8517	0.8229	0.8673	0.8673	0.8607	0.8751	0.8771	0.8842	0.8829	0.8517	0.8673
Chol	0.8709	0.9024	0.9170	0.9170	0.9106	0.8753	0.8808	0.9063	0.8979	0.8709	0.9170

^1^—log P and log D models prediction by Marvin Sketch ver. 19.9 software [36]. ^2^—log P model prediction by ALOGPS ver. 2.1 [37]. ^3^—log P models calculated by ChemDraw Ultra 10.0. software [38]. ^4^—log P calculated by MedChem Designer 3.0.0.30 tools [39].

## Data Availability

All data are available in “Appendix A” of this contribution.

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
