# Peer review of "Comparison of HPLC, HPTLC, and In Silico Lipophilicity Parameters Determined for 5-Heterocyclic 2-(2,4-Dihydroxyphenyl)-1,3,4-thiadiazoles"

_molecules, 2024, doi:10.3390/molecules29112478_

Round 1

Reviewer 1 Report

Comments and Suggestions for Authors

The authors have designed the study based on their expertise and they have done their best into this. This data is suitable to publish but I have some suggestions for the authors to improve more quality of the manuscript.

1. The first thing authors only did was In-silico approaches to their compounds, I think it would be better to add any biological activities to strengthen the data.

2. In computational experiments have the authors compared results with any standard?

3. What was the reason to use Marvin Sketch software? Why not others?

4. Are there any changes in the results with different software analyses?

Comments on the Quality of English Language

English is good and well-written. Just need to overlook some typos errors.

Author Response

Replies to Reviewer 1 remarks:    

1. The compounds have not been tested for biological effects yet and we do not have complete data on their biological potency. They were obtained as compounds with potential antiproliferative effects on cancer cells and cholinesterases inhibitors. Only two compounds from this group have been evaluated  for activity against cholinesterases so far (ref. 31. Skrzypek, A. et al. Biological evaluation and molecular docking of novel 1,3,4-thiadiazole-resorcinol conjugates as multifunctional cholinesterases inhibitors. Bioorg Chem 2021, 107, 104617)

2. The results were not compared to any standard, only to each other. This is a very valuable remark. We will take this remark into consideration when caring out similar research in the future.

3. We used several software with different algorithms: Marvin Sketch ver. 19.9, ALOGPS ver. 2.1, ChemDraw Ultra 10.0. and MedChem Designer. Marvin  was chosen because it allows the calculation of log D values depending on pH. Since some mobile  phases were buffered in HPLC chromatography, it seemed interesting to compare log kw for pH 7.4 mobile phase and log D7.4. The following  statement was added in the manuscript: Marvin Sketch software was included in the studies because of the ability to calculate log D a depending on pH.

4. The chromatographic lipophilicity parameters (log kw and RMw) are differentially correlated with the calculated log P values. The obtained results indicate that the calculated Clog P, AlOGP, log P and log D7.4 descriptors correlate the best with the chromatographic data. This depends not so much on the software (some contain several algorithms), but on the used counting method used. Some softwares contain the same counting methods, e.g. Clog or MlogP calculations and  they give the same results.

5. The manuscript was checked by a colleague fluent in English writing.

Thank you very much for all valuable remarks which will surely contribute to better understanding of the research topic by prospective readers.

Reviewer 2 Report

Comments and Suggestions for Authors

The manuscript contains a very detailed analysis of chromatographic and calculated lipophilicity data of compounds of potential biological activity; lipophilicity is, indeed, an important parameter governing biological activity of solutes as well as their behavior in the environment so such analysis (including the relationships between lipophilicity parameters obtained by different procedures) is always of interest.

In my opinion, however, this paper lack some important, yet easily accessible features. The Authors consider lipophilicity to be the main parameter distinguishing drug-like compounds (in the sense of Lipinski's Ro5 and other similar rules) from those that are not drug-like and this is, of course, correct. However, there are also other parameters (that can be easily calculated, even with open source software) that are related to drug-likeness, e.g.  Polar Surface Area, the total count of H-bond donors and acceptors, molecular weight etc. My recommendation is that the Authors should include and discuss them as well -  this could be done with ease and increase the value of the manuscript significantly.

Apart from this recommendation there are some minor details, e.g.:

Eq. 1 - a typographic error - an unnecessary ")"

Lines 193-194 - "The lipophilicity of the considered compounds expressed as log P was examined in terms of determining similarity of the molecules to the drug in this respect." - I guess the Authors mean drug-likeness? This sentence is a bit clumsy, please revise

Comments on the Quality of English Language

Minor errors to be corrected, otherwise it's fine.

Author Response

Replies to Reviewer 2 remarks:

  1. When preparing the manuscript, the authors took into account the possibility of calculating parameters other than lipophilicity, which are related to drug-likeness. However, taking into account the title and purpose of the paper, these calculations were abandoned. In this respect, only the lipophilicity parameter, which is the subject of the research, was analyzed on that score. Determining the other parameters, included  e.g. in the Lipiński's rule of five, is not a problem. If necessary, they can be calculated, but in the authors' opinion this seems to be inconsistent and goes beyond the topic and purpose of the paper.

2. Eq. 1 - ")" was delated

3. The sentence was revised as: The lipophilicity level of the compounds under consideration expressed as log P was analyzed with regard to their drug-likeness.

Thank you very much for all valuable remarks which will surely contribute to better understanding of the research topic by prospective readers.

Reviewer 3 Report

Comments and Suggestions for Authors

The work is well structured and the bibliographic references are consistent with the content of the text. Therefore, this reviewer suggests that this study is potentially suitable for publication in Molecules. The manuscript is acceptable for publication if some minor revisions are made to improve the quality of the content and writing. Some comments on the content are:

Line 18. Insert "RP HPLC" in brackets.

Line 28. Keywords changed to avoid repetition with the title. Add different keywords.

Line 101. The table 1 is unclear. I cannot understand which substituents are present in the described molecules. For example, do molecules 4 and 5 have the same substituent R1? Please add margins or improve the layout of the table content.

Line 106. Equation 1: There is a closing parenthesis without a corresponding opening parenthesis. Please double-check if the formula is correct.

Line 188. Table 4. Compound 9 has two values highlighted in red. Why?

Lines 188, 189. Remove the spaces between the words. Check throughout the text. Example in “values predicted”.

Line 206. Table 5. After the table, include the meaning of the symbols present in the table as done for Table 4 (Line 189).

Line 282. Have all the molecules studied here been synthesized and characterized in the two cited works? If there are any new molecules, include their characterization (NMR - HRMS).

Line 289. Convert mg/cm3 in mg/ml.

Line 310. “The compounds were detected under UV light at 254, 280 or 330 nm at 310 room temperature.” In this sentence, are you referring to the temperature of the chromatography column? Please elaborate on this concept.

Lines 314, 319, 325, 333. In the features regarding column geometry, add L x ID. For example: “250 × 4.6 mm L. x I.D.”

Line 333. Insert the particle size value before the column geometry.

Supplementary Materials: Editing the supplementary data without interrupting the tables. Therefore, insert one table per page. Check in the table captions that the parentheses are opened and closed.

Author Response

Replies to Reviewer 3 remarks:

  1. Line 18. Insert "RP HPLC" in brackets.

The brackets were added.

  1. 2. Line 28. Keywords changed to avoid repetition with the title. Add different keywords.

There were included the following keywords: liquid chromatography, log kw, RMw instead of: lipophilicity, HPLC, HPRLC

  1. Line 101. The table 1 is unclear. I cannot understand which substituents are present in the described molecules. For example, do molecules 4 and 5 have the same substituent R1? Please add margins or improve the layout of the table content.

The table has been corrected to improve its readability. Compounds 4 and 5 have different substituent R1 - H and -Et respectively.

  1. Line 106. Equation 1: There is a closing parenthesis without a corresponding opening parenthesis. Please double-check if the formula is correct.

Eq. 1 - ")" was delated as being unnecessary.

  1. Line 188. Table 4. Compound 9 has two values highlighted in red. Why?

It was a mistake. The highlight has been removed.

  1. Lines 188, 189. Remove the spaces between the words. Check throughout the text. Example in “values predicted”.

The whole manuscript has been checked and redundant spaces have been removed.

  1. Line 206. Table 5. After the table, include the meaning of the symbols present in the table as done for Table 4 (Line 189).

The symbols in Table 5 were explained: 1 - log P and log D models prediction by Marvin Sketch ver. 19.9 software [37]. 2 - log P model prediction by ALOGPS ver. 2.1 [38]. 3 - log P models calculated by ChemDraw Ultra 10.0. software [39]. 4 - log P calculated by MedChem Designer 3.0.0.30 tools [40].

  1. Line 282. Have all the molecules studied here been synthesized and characterized in the two cited works? If there are any new molecules, include their characterization (NMR - HRMS).

Only for some compounds have been described and published the spectroscopic data IR, NMR (1H, 13C) as well as mass spectrometry MS in the cited reference (ref. 31. Skrzypek, A. et al. Biological evaluation and molecular docking of novel 1,3,4-thiadiazole-resorcinol conjugates as multifunctional cholinesterases inhibitors. Bioorg Chem 2021, 107, 104617). Most compounds will still be evaluated in terms of biological activity. Their biological activity and spectroscopic as well as MS characteristics will be published together.

  1. Line 289. Convert mg/cm3 in mg/ml.

The unit was expressed as  mg/ml.

  1. Line 310. “The compounds were detected under UV light at 254, 280 or 330 nm at 310 room temperature.” In this sentence, are you referring to the temperature of the chromatography column? Please elaborate on this concept.

The sentence was corrected as: Measurements were carried out at room temperature. The compounds were detected under UV light at 254, 280 or 330 nm.

  1. Lines 314, 319, 325, 333. In the features regarding column geometry, add L x ID. For example: “250 × 4.6 mm L. x I.D.”

Column geometry description was checked and described according to the recommendation.

  1. Line 333. Insert the particle size value before the column geometry.

The particle size value was inserted before the column geometry: 100 Å, 4 µm, 150 × 2.1 mm

  1. Supplementary Materials: Editing the supplementary data without interrupting the tables. Therefore, insert one table per page. Check in the table captions that the parentheses are opened and closed.

Supplementary Materials were corrected. One table was inserted per page. The table captions were checked and corrected.

Thank you very much for all valuable remarks which will surely contribute to better understanding of the research topic by prospective readers.